Comparative analysis of Hymenasplenium (Aspleniaceae) chloroplast genomes from China

Chang Yanfen 1 2 changyf2018@126.com
Wang Zhixin 1 2
Zhang Guocheng 1 2
Wang Na 1 2
Cao Limin 1 2
1 College of Life Sciences, Hengyang Normal University , Hengyang, Hunan , China
2 Hunan Key Laboratory for Conservation and Utilization of Biological Resources in the Nanyue Mountainous Region , Hengyang, Hunan , China
Gelfand Mikhail
Electronic publication date: 2024 Dec 18
Publication date: 2024
Volume: 12
Electronic Location ID: e18667
Received 2024 Sep 6; Accepted 2024 Nov 18
Copyright: © 2024 Chang et al.
Copyright year: 2024
Copyright holder: Chang et al.
License: This is an open access article distributed under the terms of the Creative Commons Attribution License, which permits unrestricted use, distribution, reproduction and adaptation in any medium and for any purpose provided that it is properly attributed. For attribution, the original author(s), title, publication source (PeerJ) and either DOI or URL of the article must be cited.
License URL: https://creativecommons.org/licenses/by/4.0/

Keywords: Hymenasplenium, Chloroplast genome, Hypervariable regions, Phylogenomics, Relationship

Funding: National Natural Science Foundation of China 31500171 Natural Science Foundation of Hunan Province, China 2022JJ30097, 2022JJ40012 Scientific Research Fund of Hunan Provincial Education Department, China 20A067, 21B0642 This work was supported by the National Natural Science Foundation of China (No. 31500171), the Natural Science Foundation of Hunan Province, China (Nos. 2022JJ30097 and 2022JJ40012) and the Scientific Research Fund of Hunan Provincial Education Department, China (Nos. 20A067 and 21B0642). The funders had no role in study design, data collection and analysis, decision to publish, or preparation of the manuscript.

==============================
Hymenasplenium is one of the two genera in the large fern family Aspleniaceae. A previous study explored the molecular phylogeny of this genus using several chloroplast DNA fragments and identified three major clades, one of which is the monophyletic Old World clade with southwestern China as its diversity center. To date, there were only a few studies conducted on chloroplast genomes in Hymenasplenium or Aspleniaceae, limiting the understanding of the plastome features and its role in evolution of this group. Here, we studied the complete chloroplast genomes of 12 Hymenasplenium species covering all four subclades of the Old World clade distributed in China. The length of the Hymenasplenium plastomes ranged from 151,617 to 151,930 bp, and contained 129 genes in total, comprising 87 protein-coding, 34 tRNA, and eight rRNA genes. The GC content ranged from 41.8% to 42.1%. Comparative analyses of the Hymenasplenium chloroplast genomes displayed conserved genomic structure and identical gene arrangement. A total of 1,375 simple sequence repeats and 1,639 large repeats were detected. In addition, we detailed hypervariable regions that can be helpful for further phylogenetic research and species delimitation in Hymenasplenium. Furthermore, we supported phylogenetic relationships among major groups as well as possible cryptic speciation found in previous research in the genus. Our study provides new insights into evolutionary history and basic resources for phylogenetic and taxonomic studies of the genus Hymenasplenium.

Introduction

The genus Hymenasplenium Hayata is one of the two genera in the species-rich and nearly globally distributed fern family Aspleniaceae (The Pteridophyte Phylogeny Group I, 2016; Xu et al., 2020). Unlike the large genus Asplenium L., which has more than 700 species, Hymenasplenium contains only about 60 species (The Pteridophyte Phylogeny Group I, 2016; Xu et al., 2020). Most species of Hymenasplenium are distributed in southwestern China, with some outliers extending to the tropical regions of the New World, Afro-Madagascar, and tropical Asia (Xu et al., 2018). Morphologically, members of this genus can be easily recognized by distinguishing characteristics, such as long-creeping rhizomes and mostly 1-pinnate laminae (Xu et al., 2018).

Three major clades including several subclades were identified in a previous phylogenetic study of Hymenasplenium using six chloroplast DNA fragments (Xu et al., 2018). One of the major clades is the monophyletic Old World clade with southwestern China as its diversity center. Some species in the subclades of this paleotropical clade are highly polyphyletic or paraphyletic, and thus have been regarded as complexes or aggregates, including H. excisum (C. Presl) S. Lindsay, H. hondoense (N. Murak. & Hatan.) Nakaike, H. cheilosorum (Kunze ex Mett.) Tagawa, H. unilaterale (Lam.) Hayata, and H. obliquissimum (Hayata) Sugimoto (Ching, 1965; Murakami, 1995; Lin & Viane, 2013; Xu et al., 2018; Zhang et al., 2021; Chang et al., 2022). Species delimitation and relationships within these complexes or aggregates remain obscure. Moreover, recent phylogenetic studies have showed incongruences between the plastid and nuclear topologies, suggesting hybridization and reticulate evolution, leading to confusion about the evolutionary history of Hymenasplenium (Chang et al., 2018; Zhang et al., 2021). However, there were only a small number of nuclear gene or plastid fragments that were used for the above mentioned phylogenetic studies, which generally resulted in poor resolution of phylogenetic trees. Thus, more suitable and informative polymorphic regions are needed to be identified for further investigation of species delimitation and evolution of the genus Hymenasplenium.

The whole chloroplast genomes usually contain a larger number of phylogenetic information with a sufficient mutation rate for species delimitation and phylogenetic inference than short DNA sequences (Moore et al., 2010; Zhang et al., 2017). Therefore, plastome sequences are useful for generating phylogenetic trees with robust support, particularly for those taxa in low taxonomic levels that normally only have weak support in traditional sequencing analyses (Mower & Vickrey, 2018). In recent years, an increasing number of comparative analyses of complete chloroplast genomes have been used for phylogenetic research in many land plants including ferns (Wolf & Karol, 2012; Lu et al., 2015; Wei et al., 2017; Li et al., 2018). However, a few chloroplast genomes have been reported for Hymenasplenium or Aspleniaceae species. To date, only one plastome sequence of a Hymenasplenium species is available in NCBI GenBank database (H. laterepens N. Murak. & X. Cheng ex Y. Fen Chang & K. Hori, accession number NC035856) (Wei et al., 2017), and there are no comparative chloroplast genome analyses of Hymenasplenium yet.

In this study, in order to investigate the chloroplast genome features of Hymenasplenium species and enrich the information of its complicated evolutionary history, we sequenced and assembled plastomes from species of the monophyletic Old World clade in the genus, with sampling covering all four subclades distributed in southwestern China where have proved to be the most diverse place for the clade. We aimed (1) to investigate the structural characteristics and compositional variations in Hymenasplenium chloroplast genomes from China, (2) to identify highly variable regions that can be useful for phylogenetic study and species delimitation in the genus, and (3) to construct phylogenetic trees for Hymenasplenium based on the whole chloroplast genomes. The results of this study will improve our understanding of chloroplast genome features and their contribution to species delimitation and evolution in the genus Hymenasplenium.

Materials and methods

Taxon sampling, DNA extraction, and sequencing

A total of 22 samples of 12 species from four subclades (H. hondoense subclade, H. excisum subclade, H. cheilosorum subclade and H. obliquissimum subclade) in the Old World clade of Hymenasplenium were collected from China in their native ranges (Table 1). For DNA extraction, the fresh leaf tissues of all samples were collected and properly stored, and voucher specimens were preserved in the herbarium at Hengyang Normal University (HYNU). In addition, one publicly available complete chloroplast genome of Hymenasplenium (H. laterepens, accession number NC035856), was downloaded with annotations from NCBI GenBank.

Table 1 Samples information and accession numbers that were investigated in the present study.

Species	Subclade	Location	Voucher no.	Accession number	
H. laterepens	H. hondoense subclade	Guizhou	7191	NC035856	
H. laterepens	H. hondoense subclade	Xishuangbanna, Yunnan	chang1156	PQ187905	
H. apogamum	H. hondoense subclade	Xishuangbanna, Yunnan	chang995	PQ187899	
H. apogamum	H. hondoense subclade	Xishuangbanna, Yunnan	chang1040	PQ187901	
H. apogamum	H. hondoense subclade	Xishuangbanna, Yunnan	chang1037	PQ187903	
H. apogamum	H. hondoense subclade	Xishuangbanna, Yunnan	chang1155	PQ187904	
H. apogamum	H. hondoense subclade	Xishuangbanna, Yunnan	chang1354	PQ187915	
H. chingii	H. hondoense subclade	Gongshan, Yunnan	chang1082	PQ187898	
H. chingii	H. hondoense subclade	Miyi, Sichuan	chang1144	PQ187913	
H. wangpeishanii	H. hondoense subclade	Zhangjiajie, Hunan	chang1341	PQ187902	
H. wangpeishanii	H. hondoense subclade	Nanyue, Hunan	chang1337	PQ187906	
H. murakami-hatanakae	H. hondoense subclade	Taiwan	chencc1089	PQ187900	
H. cheilosorum	H. cheilosorum subclade	Xishuangbanna, Yunnan	chang994	PQ187896	
H. cheilosorum	H. cheilosorum subclade	Xishuangbanna, Yunnan	chang1272	PQ187909	
H. cheilosorum	H. cheilosorum subclade	Xishuangbanna, Yunnan	chang1280	PQ187910	
H. excisum	H. excisum subclade	Xishuangbanna, Yunnan	chang992	PQ187897	
H. excisum	H. excisum subclade	Xishuangbanna, Yunnan	chang1045	PQ187914	
H. obtusidentatum	H. excisum subclade	Xishuangbanna, Yunnan	chang1258	PQ187907	
H. obtusidentatum	H. excisum subclade	Xishuangbanna, Yunnan	chang1261	PQ187908	
H. furfuraceum	H. obliquissimum subclade	Gongshan, Yunnan	chang1072	PQ187895	
H. sinense	H. obliquissimum subclade	Jinping, Yunnan	chang1108	PQ187911	
H. wuliangshanense	H. obliquissimum subclade	Jingdong, Yunnan	chang1102	PQ187894	
H. retusulum	H. obliquissimum subclade	Yongde, Yunnan	chang1064	PQ187912	

Using a Plant Genomic DNA Kit (Tiangen Biotech, Beijing, China), the total genomic DNA was extracted following the manufacturer’s protocol. A Qubit 2.0 (Life Technologies, Carlsbad, CA, USA) was used to quantify the extracted DNA. The qualified DNA (≥50 ng) was then sequenced on a HiSeq platform (Illumina, San Diego, CA, USA), and a paired-end (150 bp) DNA library was constructed.

Plastome sequence assembly and annotation

To remove low-quality reads from the sequenced raw data, the fastp v.0.23.1 software (Chen et al., 2018) was used to filter the raw reads. NOVOPlasty 3.7 (Dierckxsens, Mardulyn & Smits, 2016) pipeline was used to assemble the filtered reads of genome-related chloroplasts, with the ribulose 1,5-bisphosphate carboxylase/oxygenase (rbcL) gene from H. laterepens (accession number KY427350) used as a seed sequence, and the chloroplast genome sequence of H. laterepens (accession number NC035856) used as the reference genome. After aligning with the reference genome sequence, Geneious Prime 2024.0 (Kearse et al., 2012) was used to annotate the plastome sequences of Hymenasplenium, with necessary manual adjustments. The assembled and annotated plastome sequences were all deposited in NCBI GenBank (see Table 1 for accession numbers). OrganellarGenomeDRAW (OGDRAW) (Greiner, Lehwark & Bock, 2019) was then used to visualize the circular chloroplast genome maps.

Chloroplast genome comparative analyses

MAFFT v.7 (Katoh & Standley, 2013), with default parameters, was performed to align the 22 newly sequenced chloroplast genome sequences, along with the single sequence downloaded from NCBI GenBank (H. laterepens, accession number NC035856). The trimAI v1.2 (Capella-Gutiérrez, Silla-Martinez & Gabaldón, 2009) was then performed for the alignment for further trimming, with the “-gappyout” setting used. The IRscope online tool (Amiryousefi, Hyvönen & Poczai, 2018) was used to visualize expansions and contractions of IR regions. DnaSP v.5 (Librado & Rozas, 2009) was used to calculate the nucleotide diversity (Pi), with a window length set to 800 bp and a step size set to 200 bp. The mVISTA online tool (Frazer et al., 2004) in Shufe-LAGAN mode was used to analyze the plastome divergence of all 23 Hymenasplenium chloroplast genomes, using H. laterepens (accession number NC035856) as a reference.

Repeat sequence identification

The REPuter online tool (Kurtz et al., 2001) was used to look for the short-dispersed repeats (SDRs) of the 23 Hymenasplenium chloroplast genome sequences. With the cutoff point set to ≥30 bp, sequence identities set to 90%, minimum repeat size set to 30 bp, and Hamming distance value set to three, the palindromic, forward, complementary, and reverse repeats were then identified. To analyze the tandem repeat sequences, a Tandem Repeat Finder (Benson, 1999) was used, with default parameters. The MIcroSAtellite (MISA) identification tool (Beier et al., 2017) was applied to identify simple sequence repeats (SSRs), with 10, five, four, three, three, and three selected for the minimum number of SSRs for mono-, di-, tri-, tetra-, penta-, and hexanucleotides, respectively.

Phylogenomic analysis

To investigate the relationships between the 12 Hymenasplenium species, we performed phylogenomic analyses with the 23 complete plastome sequences. Based on the study of Xu et al. (2020), Asplenium pekinense Hance, A. komarovii Akasswa and A. scolopendrium L. from closely related genus in Aspleniaceae with complete chloroplast genomes available from GenBank, were selected as the outgroup. MAFFT v.7 (Katoh & Standley, 2013) was used to align all sequences, and trimAI v1.2 (Capella-Gutiérrez, Silla-Martinez & Gabaldón, 2009) was used to trim the alignment, with the “-gappyout” setting used.

Maximum likelihood (ML) and Bayesian inference (BI) were conducted for phylogenetic analyses for Hymenasplenium. IQ-TREE (Nguyen et al., 2015) was used to reconstruct the ML tree, before that, ModelFinder (Kalyaanamoorthy et al., 2017) was used to select the best-fit nucleotide substitution model. The supports of ML tree branches were evaluated using 10,000 ultrafast bootstrap (UFBS) replicates (Minh, Nguyen & Von Haeseler, 2013). BI analysis was performed using MrBayes version 3.2 (Ronquist et al., 2012), with 2,000,000 generations under the GTR + F + I + G4 model and four chains (one cold and three heated). Run convergence was accepted when the average standard deviation (d) of split frequencies was below 0.01. In order to construct consensus trees in BI analysis, the first 25% of the trees were discarded as burn-in, and then the remaining trees were applied to build majority-rule consensus trees. FigTree v.1.4.2 (Rambaut, 2012) was used to visualize the final trees from both ML and BI analyses.

Results

Structure and features of Hymenasplenium plastomes

The size of the Hymenasplenium plastomes ranged from 151,617 (chang994: H. cheilosorum) to 151,930 bp (chang995: H. apogamum (N. Murak. & Hatan.) Nakaike) (Table 2), and showed a typical quadripartite structure, consisting of two single-copy regions (LSC and SSC) and a pair of inverted repeats (IRa and IRb) (Fig. 1). The length of the LSC region ranged from 83,366 (chang994: H. cheilosorum) to 83,554 bp (chencc1089: H. murakami-hatanakae Nakaike), and that of the SSC region ranged from 21,249 (chang1272: H. cheilosorum) to 21,394 bp (chang1072: H. furfuraceum (Ching) Viane & S. Y. Dong) (Table 2). The length of the IR regions ranged from 23,453 (chang1072: H. furfuraceum) to 23,558 bp (chang995: H. apogamum). Furthermore, the GC content of the genomes varied in a range of 41.8% to 42.1% (Table 2). All chloroplast genomes of Hymenasplenium shared the same gene arrangement and order, and kept the same number of genes, containing 129 genes, including 87 protein-coding, 34 tRNA, and eight rRNA genes (Table 2).

Table 2 Summary of the 23 complete chloroplast genomes of Hymenasplenium.

Species	Sample/voucher No.	Size (bp)	LSC (bp)	SSC (bp)	IR (bp)	Number of protein-coding genes	Number of tRNA genes	Number of rRNA genes	GC content (%)	
H. laterepens	NC035856	151,723	83,380	21,299	23,522	87	34	8	42.0	
H. laterepens	chang1156	151,725	83,388	21,295	23,521	87	34	8	42.0	
H. apogamum	chang995	151,930	83,504	21,310	23,558	87	34	8	42.0	
H. apogamum	chang1040	151,902	83,506	21,310	23,543	87	34	8	42.0	
H. apogamum	chang1037	151,906	83,510	21,308	23,544	87	34	8	42.0	
H. apogamum	chang1155	151,923	83,522	21,305	23,548	87	34	8	42.0	
H. apogamum	chang1354	151,904	83,510	21,308	23,543	87	34	8	42.0	
H. chingii	chang1082	151,742	83,457	21,272	23,505	87	34	8	41.8	
H. chingii	chang1144	151,767	83,486	21,271	23,505	87	34	8	41.9	
H. wangpeishanii	chang1341	151,674	83,475	21,271	23,464	87	34	8	41.8	
H. wangpeishanii	chang1337	151,676	83,475	21,269	23,466	87	34	8	41.8	
H. murakami-hatanakae	chencc1089	151,892	83,554	21,298	23,520	87	34	8	41.8	
H. cheilosorum	chang994	151,617	83,366	21,255	23,498	87	34	8	42.1	
H. cheilosorum	chang1272	151,631	83,368	21,249	23,507	87	34	8	42.1	
H. cheilosorum	chang1280	151,765	83,439	21,276	23,525	87	34	8	42.1	
H. excisum	chang992	151,779	83,523	21,296	23,480	87	34	8	42.0	
H. excisum	chang1045	151,867	83,531	21,298	23,519	87	34	8	42.0	
H. obtusidentatum	chang1258	151,798	83,455	21,269	23,537	87	34	8	42.0	
H. obtusidentatum	chang1261	151,793	83,463	21,256	23,537	87	34	8	42.1	
H. furfuraceum	chang1072	151,707	83,407	21,394	23,453	87	34	8	42.1	
H. sinense	chang1108	151,734	83,367	21,317	23,525	87	34	8	42.1	
H. wuliangshanense	chang1102	151,778	83,410	21,293	23,538	87	34	8	42.1	
H. retusulum	chang1064	151,840	83,431	21,295	23,557	87	34	8	42.1	

Figure 1 Representative chloroplast genome of Hymenasplenium.

The coloured bars indicate different functional groups. The thick lines of the large circle indicate the extent of the inverted repeat regions (IRa and IRb), which separate the genome into small (SSC) and large (LSC) single-copy regions. Genes shown on the outside of the circle are transcribed counterclockwise, and genes inside are transcribed clockwise. The darker grey in the inner circle corresponds to the GC content, and the lighter grey corresponds to the AT content.

Hypervariable region in Hymenasplenium species

The mutational hotspot regions among Hymenasplenium chloroplast genomes were revealed by aligning and comparing of the plastome sequences using the mVISTA online tool. The results suggested that the chloroplast genomes of Hymenasplenium were highly conserved (Fig. S1). However, hypervariable regions were also detected, including matK-rps16, chlB-trnR(UCU), rpoB-trnD(GUC), psbM-petN, trnG(GCC)-trnS(UGA), psbD-trnT(GGU), trnL(UAA)-trnF(GAA), petA-psbL, psbE-petL, rps8-rpl14, rpl16-rps23, trnT(UGU)-trnR(ACG), rpl32-trnP(GGG), trnP(GGG)-trnL(UAG), ccsA-ndhD, rps15-ycf1, and trnN(GUU)-ycf2 (Fig. S1). These regions mainly distributed in intronic and intergenic parts of the genome. Still, in protein-coding regions, a few variabilities were also detected, such as matK, rpoC2, and ycf1 (Fig. S1).

Analyses of sequence divergence by DnaSP showed that the nucleotide diversity (π) of the Hymenasplenium plastomes is from 0.0 to 0.00923. With a cutoff value of π ≥ 0.007, we observed six highly variable regions in non-coding regions: three in the LSC region (matK-rps16, chlB-trnR(UCU) and psbM-petN), and the other three in the SSC region (rpl32-trnP(GGG), trnP(GGG)-trnL(UAG), and rps15-ycf1) (Fig. 2). No highly variable regions were identified in the IR regions. The π values in the IR regions were lower than that of the LSC and SSC regions, which were all below 0.0035.

Figure 2 Nucleotide diversity (π) by sliding window analysis in the multiple alignments of the 23 Hymenasplenium plastomes.

Window length: 800 bp, step size: 200 bp. X-axis: the position of the midpoint of a window. Y-axis: the nucleotide diversity of each window.

Expansion and contraction of the IR regions

Analysis of the comparison of IR/SC borders showed that the gene arrangements and contents of Hymenasplenium chloroplast genomes were highly conserved (Fig. 3). Lengths of the IR regions were relatively consistent among all samples analyzed. The LSC/IRb border of all plastomes had a distance of 83–91 bp from the trnI gene. The junction of SSC/IRb was located in ndhF gene, but extended into the IRb region with a range from 40 to 41 bp, except for chang1072 (H. furfuraceum), which had a distance of 12 bp away from the ndhF gene. The junction of IRa/SSC was located in the chlL gene, but extended into the IRa region with a range from 54 to 55 bp, except for chang1072 (H. furfuraceum) too, which had a distance of three base pairs away from the chlL gene. The LSC/IRa junctions of all analyzed Hymenasplenium chloroplast genomes had a distance of 737–738 bp away from the trnT gene.

Figure 3 Comparison of the SC/IR junctions among the 23 Hymenasplenium plastomes.

The figure shows the gene arrangements and contents of Hymenasplenium chloroplast genomes were highly conserved except for chang1072 (H. furfuraceum) in the junctions of IRa/SSC and SSC/IRb. JLB, LSC/IRb boundary; JSB, SSC/IRb boundary: JSA, SSC/IRa boundary; and JLA, LSC/IRa boundary.

SSR polymorphisms and long repeat sequence analysis

Using MISA identification tool, 1,375 SSRs of five types (mononucleotides, dinucleotides, trinucleotides, tetranucleotides, and compound repeats) were identified among the 23 Hymenasplenium chloroplast genomes (Fig. 4). The total number of SSRs ranged from 50 (chang1064 H. retusulum) to 64 (chang1354 H. apogamum). The most abundant SSRs were the mononucleotides, primarily consisting of A and T bases. The second-most abundant SSRs were the dinucleotides, primarily consisting of AT and TA bases. The third-most abundant SSRs were the compound repeats. There were only a small number of trinucleotides and tetranucleotides that had been detected.

Figure 4 Analysis of simple sequence repeat (SSR) types detected in the 23 Hymenasplenium plastomes showing the number of mononucleotides, dinucleotides, trinucleotides, tetranucleotides, and compound repeats.

We identified 1,639 large repeats among the 23 samples of the Hymenasplenium plastomes, with chang1108 (H. sinense K. W. Xu, Li Bing Zhang & W. B. Liao) having the most (79), and chang1354 (H. apogamum) having the least (61) (Fig. 5). Forward, palindromic, and tandem repeats were found among all samples, but only changg1082 (H. chingii K. W. Xu, Li Bing Zhang & W. B. Liao), chang1258 (H. obtusidentatum Y. Fen Chang & G. Cheng Zhang), chang1280 (H. cheilosorum), and chang1108 (H. sinense) had complementary repeats. The following showed no reverse repeats: chang1102 (H. wuliangshanense (Ching) Viane & S. Y. Dong), chang1072 (H. furfuraceum), chang1108 (H. sinense), and chang1064 (H. retusulum) (Fig. 5).

Figure 5 Analysis of long repeats in the 23 Hymenasplenium plastomes showing the number of complementary, forward, palindromic, reverse, and tandem long repeats.

Phylogenomic analyses

We reconstructed phylogenetic trees for the 12 Hymenasplenium species using both ML and BI methods, based on a 154,861 bp long alignment of plastome sequences. Phylogenetic trees generated from ML and BI analyses displayed identical topology, all with significant bootstrap support (BS) and posterior probabilities (Fig. 6; ML BS: 100%; BI PP: 1.00). Hymenasplenium was resolved as monophyletic, and relationships of interspecific lineages were well resolved.

Figure 6 Maximum likelihood phylogeny based on complete chloroplast genome sequences.

Maximum parsimony and Bayesian analyses recovered identical topologies with respect to the relationships among the main clades of the paleotropical Hymenasplenium. Numbers near the nodes are ML bootstrap support values (BS, left of the slashes) and Bayesian posterior probabilities (PP, right of the slashes). Asterisks indicate 100% BS or 1.00 PP. Columns on the right refer to the subclades described in the study of Xu et al. (2018).

Based on the phylogenetic analyses, four highly supported clades were recognized within Hymenasplenium (Fig. 6). Each of the four clades was resolved as monophyletic and corresponded to the four southwestern China distributed subclades identified by Xu et al. (2018) in the Old World clade of the genus. H. hondoense subclade comprised five species (H. laterepens, H. apogamum, H. murakami-hatanakae, H. chingii and H. wangpeishanii Li Bing Zhang & K. W. Xu). H. excisum subclade contained two species (H. obtusidentatum and H. excisum). H. obliquissimum subclade comprised four species (H. wuliangshanense, H. retusulum, H. furfuraceum, and H. sinense), and is sister to H. cheilosorum subclade, which contained only one species (H. cheilosorum).

Discussion

Conserved plastome structure and hypervariable regions in Hymenasplenium

The chloroplast genomes of the 12 Hymenasplenium species were quadripartite, a typical genome structure of most land plants (Mower & Vickrey, 2018) (Fig. 1). Specifically, the 23 plastomes analyzed here were extremely conserved because they were quite consistent across genome size, structure, gene order and content (Table 2, Fig. 3). The difference in the chloroplast genome sizes of Hymenasplenium species was only 207 bp. Expansion and contraction of the IR regions are usually associated with the plastome size variation within a genus (Ravi et al., 2007; Wicke et al., 2011). However, there were only a few variations of gene distribution and contents among all IR boundary regions in Hymenasplenium (Fig. 3), which might have led to the minor differences in plastome size in the genus.

Although the 23 chloroplast genomes of Hymenasplenium showed high conservation, several hypervariable regions were detected based on sliding window and mVISTA analyses (Fig. 2, Fig. S1). In general, as observed in other fern groups and some angiosperms (Mower & Vickrey, 2018; Luo et al., 2021; Wu et al., 2021), the non-coding and single-copy regions exhibited higher levels of divergence than that of the coding and IR regions in Hymenasplenium. High mutational regions in chloroplast genome normally offer plenty of phylogenetic information, and hence can be widely used for the delimitation of closely related plant species (Li et al., 2015; Bi et al., 2018; Zeng et al., 2018). To date, phylogenetic analyses of Hymenasplenium have been based only on a small number of plastid markers, such as the rbcL gene, the atpB gene, the trnL intron, the trnL-trnF intergenic space, the rps4 gene, the rps4-trnS intergenic space, the trnH-psbA intergenic space and the psbA gene (Murakami & Schaal, 1994; Murakami et al., 1999; Chang et al., 2018; Xu et al., 2018, 2019, 2020; Zhang et al., 2021; Qiu et al., 2022). However, none of the markers mentioned above are from the highly variable regions identified here. To better resolve problems of species delimitation and DNA barcoding in those complexes or aggregates in Hymenasplenium, it would be helpful to use the hypervariable regions detected in this study for phylogenetic analyses in the genus.

Repetitive sequence analysis

SSRs in chloroplast genomes usually contain a high rate of polymorphism and rich variation at species level (Ebert & Peakall, 2009), therefore, they can be useful molecular markers for genotyping (Yang et al., 2011; Xue, Wang & Zhou, 2012) and population genetics studies (Doorduin et al., 2011). In this work, a total of 1,375 SSR loci were detected among the 23 Hymenasplenium chloroplast genomes. These newly identified SSR loci can be widely applied in future research on phylogeography and genetic diversity at the intra/interspecific and population levels for the genus. Similar to some angiosperms (Wakasugi, Tsudzuki & Sugiura, 2001), mononucleotides (A/T) were the most abundant SSRs in Hymenasplenium plastomes (Fig. 4). The number of SSRs varied wildly among species, such as that observed in H. apogamum, H. wangpeishanii and H. excisum, which all had more than 60 SSRs, and that of the other species were all less than 60 (Fig. 4). Differences in SSRs were also noted within species, such as in H. laterepens, H. chingii and H. cheilosorum (Fig. 4). Gao et al. (2018) has found that differences of repeat structures are molecular basis in response to severe environments in Dryopteris fragrans (L.) Schott. Future studies are needed to find out if SSRs varies in Hymenasplenium are associated with its environmental adaption mechanisms.

In addition to SSRs, large repeats (lengths longer than 30 bp), which usually create insertion/deletion rearrangements and mismatches (Weng et al., 2014; Asaf et al., 2018), contain great genomic variation too. A large number of long repeats (1,639) were detected in the Hymenasplenium plastomes. As with angiosperms (Huang et al., 2019; Li et al., 2021), palindromic, forward, and tandem repeats were found to be common among these large repeats, whereas reverse and complementary repeats were rare (Fig. 5). In Hymenasplenium, specifically, all species in the four subclades had palindromic, forward, tandem and complementary repeats, but four samples from the H. obliquissimum subclade had no reverse repeats (Fig. 5). Therefore, further studies are necessary for evaluating the roles of different kinds of large repeats in exploring the evolutionary history in Hymenasplenium.

Phylogenetic inference

A well-resolved phylogeny for Hymenasplenium with limited number of species sampled was presented in this study (Fig. 6). The results strongly supported the monophyly of the four subclades identified in a previous study based on chloroplast DNA fragments (Xu et al., 2018). Relationships among species found in this study were largely consistent with previous plastid phylogenies (Xu et al., 2018), such as the closely related H. laterepens and H. apogamum (Chang et al., 2018), and the sister group of H. cheilosorum subclade and H. obliquissimum subclade (Xu et al., 2018). In addition, our phylogenetic results provided strong support for species in the H. obliquissimum subclade that were previously well supported only in BI analysis (Zhang et al., 2021). These findings supported the notion that plastomes can be good markers to construct the phylogeny of Hymenasplenium or Aspleniaceae with a larger species sampling.

Although H. cheilosorum was monophyletic, three samples included in this study were resolved into two well supported groups (Fig. 6). Our results could support the point of view of Xu et al. (2018) that H. cheilosorum contains more than one species. Similar situations existed in H. apogamum of H. hondoense subclade in which five samples were resolved into three groups (Fig. 6). The taxonomy of these groups still needs further attention to confirm if they represent distinct taxa because cryptic speciation including hybridization/polyploidization was found to be common in Hymenasplenium (Xu et al., 2018; Zhang et al., 2021). In that case, phylogenetic analyses of nuclear DNA will be necessary, because the chloroplast sequences usually possess lower mutation rate and have limitations in studying reticulate evolution. In addition, samples in H. apogamum, H. excisum and H. cheilosorum were all from the same location of Xishuangbanna, south of Yunnan province. This reinforces that southwestern China is the most diverse place of Hymenasplenium and suggests that additional detailed sampling at population levels for further taxonomy and phylogenetic study is required for the genus.

Conclusions

This study is the first attempt to comprehensively examine plastome features and infer phylogeny using plastome data for Hymenasplenium. The results revealed a highly conserved chloroplast genome structure of the genus, and displayed six hypervariable regions and a large number of repeat sequences. The genomic mutational regions identified here can be widely used as high-resolution DNA markers in future phylogeographic and phylogenetic studies in Hymenasplenium. Results of phylogenomic analyses were generally consistent with previous research, and might approve of the possibly existing cryptic speciation in the genus. For a better understanding of the complex evolutionary history of Hymenasplenium, future studies should focus on phylogenetic reconstructions based not only on complete chloroplast genomes but also on nuclear genes from genome-skimming data, and should also incorporate molecular data, as well as evidence from distribution, chromosome and morphology.

Supplemental Information

Supplemental Information 1 Chloroplast genome global comparison map of Hymenasplenium using mVISTA.

X-axis: coordinate in the chloroplast genome; Y-axis: level of variation (50%–100%). Genome regions are color-coded, distinguishing between exons (purple), introns (blue), and intergenic-spacers (IGS) (red)

Supplemental Information 2 Plastome sequences deposited in NCBI GenBank.

We thank several colleagues who assisted with the collections, including ChiChuan Chen, Huafeng Hong and Changle Zhang in China.

Additional Information and Declarations

Competing Interests

Author Contributions

Data Availability

The authors declare that they have no competing interests.

Yanfen Chang conceived and designed the experiments, analyzed the data, prepared figures and/or tables, authored or reviewed drafts of the article, and approved the final draft.

Zhixin Wang analyzed the data, prepared figures and/or tables, and approved the final draft.

Guocheng Zhang performed the experiments, prepared figures and/or tables, and approved the final draft.

Na Wang performed the experiments, prepared figures and/or tables, and approved the final draft.

Limin Cao analyzed the data, authored or reviewed drafts of the article, and approved the final draft.

The following information was supplied regarding data availability:

The plastome sequences are available at NCBI GenBank: PQ187894–PQ187915.

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
