# Peer review of "Comparative analysis of Hymenasplenium (Aspleniaceae) chloroplast genomes from China"

_PeerJ, doi:10.7717/peerj.18667_

## Round 0.1 · original submission · Major Revisions

All 3 reviewers note the importance of the study, although two of them have major concerns about interpretation of the results and a considerable number of editorial comments of ranging from relatively minor to rather serious (e.g. proper use of nomenclature).

Reviewer 1 ·

Basic reporting

This paper presents a very straightforward and well-justified case: resolving the phylogenetic relationships of a clade of the genus Hymenasplenium with more powerful techniques than those used to date. The introduction presents well the previous context and the objectives are clear. The methodology is clear and well explained, and is matched by the results, which are also very well presented.
There are some problems with scientific names. It is advisable to put the authors of the species the first time they are mentioned, but this has not been done for all cases. In the cases where it has been done, the author in parentheses of the basionym (when there is one) has been omitted (for instance, between lines 48 and 50). Also, through the text many taxa are presented without the authors the first time they are mentioned. These cases have been pointed out in the attached pdf, but some cases may have been overlooked, so I recommend that authors check that all scientific names are spelled with the authorship the first time they are mentioned and that this is done correctly. I suggest consulting IPNI for this purpose.
The information included in lines 74-75 are redundant with the first lines of material and methods (87-88). I think the details of the number of sequences and individuals sequenced is more appropriately put in Material and methods, and in the introduction simply what you want to do and why. On line 157 there is a space missing between “from” and “151,617”.
The results are clear and well explained. On line 158, in results, it is mentioned that the plastid genome structure of Hymenasplenium is similar to that of other vascular plants. However, this statement is more appropriate for discussion, and in fact it already appears there, I would remove it.
The discussion justifies very well how the results obtained in the work are useful detecting markers within the most variable plastids, with some writing problems as In line 249, when expressing “observed in other angiosperms” it could seem that we are working with angiosperms. I think it would be better to express it as “In general, as observed in other fern groups and in some angiosperms”. In line 267 it happens as in 249, by saying “similar to other angiosperms” it looks like you are working with one. I would use “similar to some angiosperms”. Also, in line 256 I'm not sure if you got the markers right, I think they are all in duplicate.
However, I see serious problems in the taxonomic interpretation of the phylogeny. Lines 292 to 294 discuss possible undescribed species that look good in the phylogeny, but I am not able to detect this, nor is it cited in which paper these putative new species are mentioned. Overall, the conclusions of this paragraph are not well understood, and I suggest that they be rewritten and reevaluated. It is similar with the following, between lines 297 and 300 it is said that certain lineages are confirmed in this work as species complexes, however it is impossible to infer this with the phylogeny presented, since all species form monophyletic clades. For example, Hymenasplenium excisum has two specimens in the phylogeny that form a monophyletic clade, therefore it cannot be stated that this constitutes a species complex at all. I suggest that the authors revise well this whole part of the discussion to make it coherent and understandable. Also, it is indicated that some specimens from Yunnan are identified as distinct taxa, but it would be interesting to indicate which ones so as not to have to constantly go to the table to consult it.
Respecting tables and figures, in Table 2, I think it would be informative to also put which taxon each plastid corresponds to, so as not to have to go check it in Table 1. In Figure 1 the yellow text in the center of the image is difficult to see, I would choose another font color to make it readable. In figure 6, the subclade H. cheilosorum has “cheilosorum” written with a capital letter, it should be lowercase. Likewise, the species Hymenasplenium excisum is misspelled throughout the figure as H. excism, this needs to be corrected.
All in all, I believe the work is very useful and represents significant advances in the understanding of the classification of the genus Hymenasplenium. However, the aspects mentioned should be revised. I include a pdf with comments on the text to complement this review.

Experimental design

No comments.

Validity of the findings

I think the results regarding the search for variable zones of the chloroplast of Hymenasplenium is valuable, but there are problems in the interpretation of the phylogeny that I have addressed in the basic report.

Additional comments

No comments.

Annotated reviews are not available for download in order to protect the identity of reviewers who chose to remain anonymous.

·

Basic reporting

For the most part, the paper is clearly written, particularly the introduction and methods though there are some exceptions which I detail below. Briefly, ambiguous use of ‘lineages’ and sections on cryptic species were particularly confusing. The introduction does a good job of describing previous research in this genus and laying out the gap in knowledge that this research is meant to fill. The figures were difficult to read due to text overlapping images, small fonts throughout, and generally low resolution. Specific issues with figures are noted below.

1. Throughout, your usage of the phrase ‘lineage’ is confusing (e.g. lines 221-234). I’m not sure how you define it. For instance, you say that the H. hondoense subclade is divided into 7 lineages, but this is entirely arbitrary. It would be equally correct to say that it comprises a single lineage, or two, or five… As such, it’s unclear how the ‘number of lineages’ in a given clade is relevant here. As another example, why is it relevant that H. apogamum is divided into three lineages? A single species can contain multiple lineages, a genus can be considered a single lineage. The term is deliberately non-specific. Because of this you need to provide context that gives it meaning such as ‘highly supported lineages’ if you have variation in support values. Or maybe ‘distantly related’ or ‘deeply divergent’ lineages if they are on long branches.
2. Line 273: I don’t know of any studies using chloroplast SSRs to conduct population genetics analysis in ferns. That is not to say they don't exist but the cited article (Gao et al. 2018) does not include any population level statistics as it only presents a single cp genome for a single species (though it does compare repetitive sequence content with other published fern chloroplast genomes).
3. Lines 292-293 “… those possibly undescribed species in the H. hondoense subclade…” which species and which samples are you referring to? Also, what does possibly undescribed species mean here?
4. Lines 294-296: As it’s written, this feels like a sort of common knowledge statement. Using more sequence data results in better resolved phylogenies. Of course it does! Maybe rephrase to highlight the utility of chloroplast sequences specifically in this genus/family. Also, in this case, you could mention some of the limitations of chloroplast sequences, e.g. cannot be used to detect introgression, lower mutation rate.
6. Text in the center of figure 1 is basically illegible. Maybe try a different color or a semitransparent background to make the text stand out better. Alternatively you could just move the text somewhere else in the figure. Annotations are also very small and difficult to read, consider resizing them.
7. Should mention which species is represented in figure 1 in the legend.
8. Figure 3 text is also very small and difficult or impossible to read at this resolution (even zooming in). It could also use more detail in the figure legend. Maybe briefly mention what I am supposed to see here? Are you trying to show that this region is highly conserved? Are there differences that could be highlighted either in the figure or the legend?
9. Figure 4 and 5: are these samples grouped by species? It would be nice to label species on here as the sample IDs are not immediately meaningful to me.
10. Colors need to be explained in the supplemental figure.

Experimental design

Overall, the research seems to be a good fit for the journal and presents original research in an understudied group. Research questions are well stated as is the gap in knowledge that this research will fill. The methods are thoroughly described and I believe that any researcher familiar with phylogenetic analysis would be able to replicate the study using the raw data provided. All the methods described here are standard in the field and appropriate to the questions being asked. The authors generate a well-supported phylogeny that seems to corroborate species groupings from previous research.

Validity of the findings

Aside from ambiguous use of ‘lineages’ the results are clearly stated and represented appropriately by the figures. However, the discussion and conclusions begin to break down a bit. There are several non-specific references to “discordance between morphology based traditional taxonomy and molecular phylogenetics” without detailing the nature of that discordance or providing citations. The discussion of cryptic species seems unmerited and does not appear to be supported by the results. All putative species form well-supported monophyletic clades regardless of how many ‘lineages’ they comprise. I believe that the delineation of hypervariable regions is valuable but the significance is poorly stated in the discussion (specific comment below). Finally the identification of SSRs and other repeats would be greatly enhanced by an exploration of inter and intra-specific variation at each locus and not just variation in the relative number of each type of SSR.

1. Lines 301-306, I’m not sure I understand this statement or earlier references to cryptic species in the abstract. All of the species in your phylogeny (figure 6) appear to be well supported and monophyletic. I do not see any obvious evidence of cryptic species complexes or misplaced taxa here, no polyphyletic or paraphyletic species or individuals out on long branches. If your results differ substantially from those of others (e.g. using morphology or other genetic data) you should clearly and specifically mention areas of discordance. Also, as I state in an earlier comment, more chloroplast sequences will not necessarily help resolve phylogenetic discordance, especially if uncertainty is the result of reticulate evolution.
2. Repetitive sequence analysis: You do not mention whether or what proportion of SSR loci vary within species or even between species. This is important information if you are suggesting these be used for genotyping or population genetics.
3. Lines 259-260: Phrasing is a little off here. Do you mean to say that lack of variability in markers used for previous studies has hindered phylogenetic analysis?

Additional comments

Minor notes:
1. Line 45: “a couple” generally means two. The referenced study used 6 chloroplast markers. Why not just say 6?
2. Line 258: ”makers” should be “markers”
3. Line 225-226: southwest of China should be "in the southwest of china" or just "in southwest China"

In sum, I believe that this paper presents sound research and potentially provides a valuable resource for future research in the Aspleniaceae but is hindered by occasionally confusing and ambiguous language and seemingly unmerited conclusions. As such, I expect it will require significant revision prior to publication.

Reviewer 3 ·

Basic reporting

The article is well-written overall, but there are a few small spelling and grammar mistakes to be corrected.

Experimental design

no comment

Validity of the findings

no comment

Additional comments

The authors provide new chloroplast genomes to help reconstruct the evolutionary history of Hymenasplenium. This is a nice contribution to our understanding of this group, providing not just a new phylogenetic hypothesis but molecular resources for future work.

My only minor comments are that it would be nice to see a bit more discussion of how this new tree impacts the understanding of the group, and where future work would go.

---

## Round 0.2 · accepted · Accept

The reviewer suggest checking grammar, spelling errors, and typos, but aside of that the paper seems to be fine.

·

Basic reporting

The results and discussion are much improved in this revised draft. I suggest going over the draft once more for grammar, though I only found a single example where phrasing was particularly confusing:

Line 86: “where have proved to be” should be “which has proved to be.” But even this phrasing is a bit awkward. Maybe say something like: “…distributed in southwest China, the center of species diversity for this clade”

Overall, I think this revision is well written and clearly describes the background/motivation as well as key results.

Experimental design

no comment

Validity of the findings

no comment

Additional comments

no comment